# REBAR: Low-variance, unbiased gradient estimates for discrete latent variable models

**George Tucker[1],[*], Andriy Mnih[2], Chris J. Maddison[2],[3],**
**Dieterich Lawson[1],[*], Jascha Sohl-Dickstein[1]**
[1]Google Brain, [2]DeepMind, [3]University of Oxford
{gjt, amnih, dieterichl, jaschasd}@google.com
cmaddis@stats.ox.ac.uk

## Abstract

Learning in models with discrete latent variables is challenging due to high variance gradient estimators. Generally, approaches have relied on control variates to reduce the variance of the REINFORCE estimator. Recent work (Jang et al., 2016; Maddison et al., 2016) has taken a different approach, introducing a continuous relaxation of discrete variables to produce low-variance, but biased, gradient estimates. In this work, we combine the two approaches through a novel control variate that produces low-variance, *unbiased* gradient estimates. Then, we introduce a modification to the continuous relaxation and show that the tightness of the relaxation can be adapted online, removing it as a hyperparameter. We show state-of-the-art variance reduction on several benchmark generative modeling tasks, generally leading to faster convergence to a better final log-likelihood.

## 1 Introduction

Models with discrete latent variables are ubiquitous in machine learning: mixture models, Markov Decision Processes in reinforcement learning (RL), generative models for structured prediction, and, recently, models with hard attention (Mnih et al., 2014) and memory networks (Zaremba & Sutskever, 2015). However, when the discrete latent variables cannot be marginalized out analytically, maximizing objectives over these models using REINFORCE-like methods (Williams, 1992) is challenging due to high-variance gradient estimates obtained from sampling. Most approaches to reducing this variance have focused on developing clever control variates (Mnih & Gregor, 2014; Titsias & Lázaro-Gredilla, 2015; Gu et al., 2015; Mnih & Rezende, 2016). Recently, Jang et al. (2016) and Maddison et al. (2016) independently introduced a novel distribution, the *Gumbel-Softmax* or *Concrete* distribution, that continuously relaxes discrete random variables. Replacing every discrete random variable in a model with a Concrete random variable results in a continuous model where the reparameterization trick is applicable (Kingma & Welling, 2013; Rezende et al., 2014). The gradients are biased with respect to the discrete model, but can be used effectively to optimize large models. The tightness of the relaxation is controlled by a temperature hyperparameter. In the low temperature limit, the gradient estimates become unbiased, but the variance of the gradient estimator diverges, so the temperature must be tuned to balance bias and variance.

We sought an estimator that is low-variance, unbiased, and does not require tuning additional hyperparameters. To construct such an estimator, we introduce a simple control variate based on the difference between the REINFORCE and the reparameterization trick gradient estimators for the relaxed model. This reduces variance, but does not outperform state-of-the-art methods on its own. Our key contribution is to show that it is possible to conditionally marginalize the control variate

---

[*]Work done as part of the Google Brain Residency Program.
Source code for experiments: github.com/tensorflow/models/tree/master/research/rebar

to significantly improve its effectiveness. We call this the REBAR gradient estimator, because it combines REINFORCE gradients with gradients of the Concrete relaxation. Next, we show that a modification to the Concrete relaxation connects REBAR to MuProp in the high temperature limit. Finally, because REBAR is unbiased for all temperatures, we show that the temperature can be optimized online to reduce variance further and relieve the burden of setting an additional hyperparameter.

In our experiments, we illustrate the potential problems inherent with biased gradient estimators on a toy problem. Then, we use REBAR to train generative sigmoid belief networks (SBNs) on the MNIST and Omniglot datasets and to train conditional generative models on MNIST. Across tasks, we show that REBAR has state-of-the-art variance reduction which translates to faster convergence and better final log-likelihoods. Although we focus on binary variables for simplicity, this work is equally applicable to categorical variables (Appendix C).

## 2 Background

For clarity, we first consider a simplified scenario. Let $b \sim \text{Bernoulli}(\theta)$ be a vector of independent binary random variables parameterized by $\theta$. We wish to maximize

$$\mathbb{E}_{p(b)} [f(b, \theta)],$$

where $f(b, \theta)$ is differentiable with respect to $b$ and $\theta$, and we suppress the dependence of $p(b)$ on $\theta$ to reduce notational clutter. This covers a wide range of discrete latent variable problems; for example, in variational inference $f(b, \theta)$ would be the stochastic variational lower bound.

Typically, this problem has been approached by gradient ascent, which requires efficiently estimating

$$\frac{d}{d\theta} \mathbb{E}_{p(b)} [f(b, \theta)] = \mathbb{E}_{p(b)} \left[ \frac{\partial f(b, \theta)}{\partial \theta} + f(b, \theta) \frac{\partial}{\partial \theta} \log p(b) \right]. \tag{1}$$

In practice, the first term can be estimated effectively with a single Monte Carlo sample, however, a naïve single sample estimator of the second term has high variance. Because the dependence of $f(b, \theta)$ on $\theta$ is straightforward to account for, to simplify exposition we assume that $f(b, \theta) = f(b)$ does not depend on $\theta$ and concentrate on the second term.

### 2.1 Variance reduction through control variates

Paisley et al. (2012); Ranganath et al. (2014); Mnih & Gregor (2014); Gu et al. (2015) show that carefully designed control variates can reduce the variance of the second term significantly. Control variates seek to reduce the variance of such estimators using closed form expectations for closely related terms. We can subtract any $c$ (random or constant) as long as we can correct the bias (see Appendix A and (Paisley et al., 2012) for a review of control variates in this context):

$$\frac{\partial}{\partial \theta} \mathbb{E}_{p(b,c)} [f(b)] = \frac{\partial}{\partial \theta} \left( \mathbb{E}_{p(b,c)} [f(b) - c] + \mathbb{E}_{p(b,c)} [c] \right) = \mathbb{E}_{p(b,c)} \left[ (f(b) - c) \frac{\partial}{\partial \theta} \log p(b) \right] + \frac{\partial}{\partial \theta} \mathbb{E}_{p(b,c)} [c]$$

For example, NVIL (Mnih & Gregor, 2014) learns a $c$ that does not depend[2] on $b$ and MuProp (Gu et al., 2015) uses a linear Taylor expansion of $f$ around $\mathbb{E}_{p(b|\theta)}[b]$. Unfortunately, even with a control variate, the term can still have high variance.

### 2.2 Continuous relaxations for discrete variables

Alternatively, following Maddison et al. (2016), we can parameterize $b$ as $b = H(z)$, where $H$ is the element-wise hard threshold function[3] and $z$ is a vector of independent Logistic random variables defined by

$$z := g(u, \theta) := \log \frac{\theta}{1 - \theta} + \log \frac{u}{1 - u},$$

where $u \sim \text{Uniform}(0, 1)$. Notably, $z$ is differentiably reparameterizable (Kingma & Welling, 2013; Rezende et al., 2014), but the discontinuous hard threshold function prevents us from using the reparameterization trick directly. Replacing all occurrences of the hard threshold function with a continuous relaxation $H(z) \approx \sigma_\lambda(z) := \sigma\left(\frac{z}{\lambda}\right) = \left(1 + \exp\left(-\frac{z}{\lambda}\right)\right)^{-1}$ however results in a reparameterizable computational graph. Thus, we can compute low-variance gradient estimates for the relaxed model that approximate the gradient for the discrete model. In summary,

$$\frac{\partial}{\partial\theta}\mathop{\mathbb{E}}_{p(b)}\left[f(b)\right] = \frac{\partial}{\partial\theta}\mathop{\mathbb{E}}_{p(z)}\left[f(H(z))\right] \approx \frac{\partial}{\partial\theta}\mathop{\mathbb{E}}_{p(z)}\left[f(\sigma_\lambda(z))\right] = \mathop{\mathbb{E}}_{p(u)}\left[\frac{\partial}{\partial\theta}f\left(\sigma_\lambda(g(u,\theta))\right)\right],$$

where $\lambda > 0$ can be thought of as a temperature that controls the tightness of the relaxation (at low temperatures, the relaxation is nearly tight). This generally results in a low-variance, but biased Monte Carlo estimator for the discrete model. As $\lambda \to 0$, the approximation becomes exact, but the variance of the Monte Carlo estimator diverges. Thus, in practice, $\lambda$ must be tuned to balance bias and variance. See Appendix C and Jang et al. (2016); Maddison et al. (2016) for the generalization to the categorical case.

## 3  REBAR

We seek a low-variance, unbiased gradient estimator. Inspired by the Concrete relaxation, our strategy will be to construct a control variate (see Appendix A for a review of control variates in this context) based on the difference between the REINFORCE gradient estimator for the relaxed model and the gradient estimator from the reparameterization trick. First, note that closely following Eq. 1

$$\mathop{\mathbb{E}}_{p(b)}\left[f(b)\frac{\partial}{\partial\theta}\log p(b)\right] = \frac{\partial}{\partial\theta}\mathop{\mathbb{E}}_{p(b)}\left[f(b)\right] = \frac{\partial}{\partial\theta}\mathop{\mathbb{E}}_{p(z)}\left[f(H(z))\right] = \mathop{\mathbb{E}}_{p(z)}\left[f(H(z))\frac{\partial}{\partial\theta}\log p(z)\right]. \quad (2)$$

The similar form of the REINFORCE gradient estimator for the relaxed model

$$\frac{\partial}{\partial\theta}\mathop{\mathbb{E}}_{p(z)}\left[f(\sigma_\lambda(z))\right] = \mathop{\mathbb{E}}_{p(z)}\left[f(\sigma_\lambda(z))\frac{\partial}{\partial\theta}\log p(z)\right] \quad (3)$$

suggests it will be strongly correlated and thus be an effective control variate. Unfortunately, the Monte Carlo gradient estimator derived from the left hand side of Eq. 2 has much lower variance than the Monte Carlo gradient estimator derived from the right hand side. This is because the left hand side can be seen as analytically performing a conditional marginalization over $z$ given $b$, which is noisily approximated by Monte Carlo samples on the right hand side (see Appendix B for details). Our key insight is that an analogous conditional marginalization can be performed for the control variate (Eq. 3),

$$\mathop{\mathbb{E}}_{p(z)}\left[f(\sigma_\lambda(z))\frac{\partial}{\partial\theta}\log p(z)\right] = \mathop{\mathbb{E}}_{p(b)}\left[\frac{\partial}{\partial\theta}\mathop{\mathbb{E}}_{p(z|b)}\left[f(\sigma_\lambda(z))\right]\right] + \mathop{\mathbb{E}}_{p(b)}\left[\mathop{\mathbb{E}}_{p(z|b)}\left[f(\sigma_\lambda(z))\right]\frac{\partial}{\partial\theta}\log p(b)\right],$$

where the first term on the right-hand side can be efficiently estimated with the reparameterization trick (see Appendix C for the details)

$$\mathop{\mathbb{E}}_{p(b)}\left[\frac{\partial}{\partial\theta}\mathop{\mathbb{E}}_{p(z|b)}\left[f(\sigma_\lambda(z))\right]\right] = \mathop{\mathbb{E}}_{p(b)}\left[\mathop{\mathbb{E}}_{p(v)}\left[\frac{\partial}{\partial\theta}f(\sigma_\lambda(\tilde{z}))\right]\right],$$

where $v \sim \text{Uniform}(0, 1)$ and $\tilde{z} \equiv \tilde{g}(v, b, \theta)$ is the differentiable reparameterization for $z|b$ (Appendix C). Therefore,

$$\mathop{\mathbb{E}}_{p(z)}\left[f(\sigma_\lambda(z))\frac{\partial}{\partial\theta}\log p(z)\right] = \mathop{\mathbb{E}}_{p(b)}\left[\mathop{\mathbb{E}}_{p(v)}\left[\frac{\partial}{\partial\theta}f(\sigma_\lambda(\tilde{z}))\right]\right] + \mathop{\mathbb{E}}_{p(b)}\left[\mathop{\mathbb{E}}_{p(z|b)}\left[f(\sigma_\lambda(z))\right]\frac{\partial}{\partial\theta}\log p(b)\right].$$

Using this to form the control variate and correcting with the reparameterization trick gradient, we arrive at

$$\frac{\partial}{\partial\theta}\mathop{\mathbb{E}}_{p(b)}\left[f(b)\right] = \mathop{\mathbb{E}}_{p(u,v)}\left[\left[f(H(z)) - \eta f(\sigma_\lambda(\tilde{z}))\right]\frac{\partial}{\partial\theta}\log p(b)\bigg|_{b=H(z)} \right.$$
$$\left. + \eta\frac{\partial}{\partial\theta}f(\sigma_\lambda(z)) - \eta\frac{\partial}{\partial\theta}f(\sigma_\lambda(\tilde{z}))\right], \quad (4)$$

where $u, v \sim \text{Uniform}(0, 1)$, $z \equiv g(u, \theta)$, $\tilde{z} \equiv \tilde{g}(v, H(z), \theta)$, and $\eta$ is a scaling on the control variate. The REBAR estimator is the single sample Monte Carlo estimator of this expectation. To reduce computation and variance, we couple $u$ and $v$ using common random numbers (Appendix G, (Owen, 2013)). We estimate $\eta$ by minimizing the variance of the Monte Carlo estimator with SGD. In Appendix D, we present an alternative derivation of REBAR that is shorter, but less intuitive.

## 3.1 Rethinking the relaxation and a connection to MuProp

Because $\sigma_\lambda(z) \to \frac{1}{2}$ as $\lambda \to \infty$, we consider an alternative relaxation

$$H(z) \approx \sigma \left( \frac{1}{\lambda} \frac{\lambda^2 + \lambda + 1}{\lambda + 1} \log \frac{\theta}{1-\theta} + \frac{1}{\lambda} \log \frac{u}{1-u} \right) = \sigma_\lambda(z_\lambda), \tag{5}$$

where $z_\lambda = \frac{\lambda^2 + \lambda + 1}{\lambda + 1} \log \frac{\theta}{1-\theta} + \log \frac{u}{1-u}$. As $\lambda \to \infty$, the relaxation converges to the mean, $\theta$, and still as $\lambda \to 0$, the relaxation becomes exact. Furthermore, as $\lambda \to \infty$, the REBAR estimator converges to MuProp without the linear term (see Appendix E). We refer to this estimator as SimpleMuProp in the results.

## 3.2 Optimizing temperature ($\lambda$)

The REBAR gradient estimator is unbiased for *any* choice of $\lambda > 0$, so we can optimize $\lambda$ to minimize the variance of the estimator without affecting its unbiasedness (similar to optimizing the dispersion coefficients in Ruiz et al. (2016)). In particular, denoting the REBAR gradient estimator by $r(\lambda)$, then

$$\frac{\partial}{\partial \lambda} \text{Var}(r(\lambda)) = \frac{\partial}{\partial \lambda} \left( \mathbb{E}\left[r(\lambda)^2\right] - \mathbb{E}\left[r(\lambda)\right]^2 \right) = \mathbb{E}\left[ 2r(\lambda) \frac{\partial r(\lambda)}{\partial \lambda} \right]$$

because $\mathbb{E}[r(\lambda)]$ does not depend on $\lambda$. The resulting expectation can be estimated with a single sample Monte Carlo estimator. This allows the tightness of the relaxation to be adapted online jointly with the optimization of the parameters and relieves the burden of choosing $\lambda$ ahead of time.

## 3.3 Multilayer stochastic networks

Suppose we have multiple layers of stochastic units (i.e., $b = \{b_1, b_2, \ldots, b_n\}$) where $p(b)$ factorizes as

$$p(b_{1:n}) = p(b_1)p(b_2|b_1)\cdots p(b_n|b_{n-1}),$$

and similarly for the underlying Logistic random variables $p(z_{1:n})$ recalling that $b_i = H(z_i)$. We can define a relaxed distribution over $z_{1:n}$ where we replace the hard threshold function $H(z)$ with a continuous relaxation $\sigma_\lambda(z)$. We refer to the relaxed distribution as $q(z_{1:n})$.

We can take advantage of the structure of $p$, by using the fact that the high variance REINFORCE term of the gradient also decomposes

$$\mathbb{E}_{p(b)}\left[ f(b) \frac{\partial}{\partial \theta} \log p(b) \right] = \sum_i \mathbb{E}_{p(b)}\left[ f(b) \frac{\partial}{\partial \theta} \log p(b_i|b_{i-1}) \right].$$

Focusing on the $i^{th}$ term, we have

$$\mathbb{E}_{p(b)}\left[ f(b) \frac{\partial}{\partial \theta} \log p(b_i|b_{i-1}) \right] = \mathbb{E}_{p(b_{1:i-1})}\left[ \mathbb{E}_{p(b_i|b_{i-1})}\left[ \mathbb{E}_{p(b_{i+1:n}|b_i)} [f(b)] \frac{\partial}{\partial \theta} \log p(b_i|b_{i-1}) \right] \right],$$

which suggests the following control variate

$$\mathbb{E}_{p(z_i|b_i,b_{i-1})}\left[ \mathbb{E}_{q(z_{i+1:n}|z_i)} [f(b_{1:i-1}, \sigma_\lambda(z_{i:n}))] \right] \frac{\partial}{\partial \theta} \log p(b_i|b_{i-1})$$

for the middle expectation. Similarly to the single layer case, we can debias the control variate with terms that are reparameterizable. Note that due to the switch between sampling from $p$ and sampling from $q$, this approach requires $n$ passes through the network (one pass per layer). We discuss alternatives that do not require multiple passes through the network in Appendix F.

### 3.4 Q-functions

Finally, we note that since the derivation of this control variate is independent of $f$, the REBAR control variate can be generalized by replacing $f$ with a learned, differentiable $Q$-function. This suggests that the REBAR control variate is applicable to RL, where it would allow a "pseudo-action"-dependent baseline. In this case, the pseudo-action would be the relaxation of the discrete output from a policy network.

## 4  Related work

Most approaches to optimizing an expectation of a function w.r.t. a discrete distribution based on samples from the distribution can be seen as applications of the REINFORCE (Williams, 1992) gradient estimator, also known as the likelihood ratio (Glynn, 1990) or score-function estimator (Fu, 2006). Following the notation from Section 2, the basic form of an estimator of this type is $(f(b) - c)\frac{\partial}{\partial\theta}\log p(b)$ where $b$ is a sample from the discrete distribution and $c$ is some quantity independent of $b$, known as a baseline. Such estimators are unbiased, but without a carefully chosen baseline their variance tends to be too high for the estimator to be useful and much work has gone into finding effective baselines.

In the context of training latent variable models, REINFORCE-like methods have been used to implement sampling-based variational inference with either fully factorized (Wingate & Weber, 2013; Ranganath et al., 2014) or structured (Mnih & Gregor, 2014; Gu et al., 2015) variational distributions. All of these involve learned baselines: from simple scalar baselines (Wingate & Weber, 2013; Ranganath et al., 2014) to nonlinear input-dependent baselines (Mnih & Gregor, 2014). MuProp (Gu et al., 2015) combines an input-dependent baseline with a first-order Taylor approximation to the function based on the corresponding mean-field network to achieve further variance reduction. REBAR is similar to MuProp in that it also uses gradient information from a proxy model to reduce the variance of a REINFORCE-like estimator. The main difference is that in our approach the proxy model is essentially the relaxed (but still stochastic) version of the model we are interested in, whereas MuProp uses the mean field version of the model as a proxy, which can behave very differently from the original model due to being completely deterministic. The relaxation we use was proposed by (Maddison et al., 2016; Jang et al., 2016) as a way of making discrete latent variable models reparameterizable, resulting in a low-variance but biased gradient estimator for the original model. REBAR on the other hand, uses the relaxation in a control variate which results in an unbiased, low-variance estimator. Alternatively, Titsias & Lázaro-Gredilla (2015) introduced local expectation gradients, a general purpose unbiased gradient estimator for models with continuous and discrete latent variables. However, it typically requires substantially more computation than other methods. Recently, a specialized REINFORCE-like method was proposed for the tighter multi-sample version of the variational bound (Burda et al., 2015) which uses a leave-out-out technique to construct per-sample baselines (Mnih & Rezende, 2016). This approach is orthogonal to ours, and we expect it to benefit from incorporating the REBAR control variate.

## 5  Experiments

As our goal was variance reduction to improve optimization, we compared our method to the state-of-the-art unbiased single-sample gradient estimators, NVIL (Mnih & Gregor, 2014) and MuProp (Gu et al., 2015), and the state-of-the-art biased single-sample gradient estimator Gumbel-Softmax/Concrete (Jang et al., 2016; Maddison et al., 2016) by measuring their progress on the training objective and the variance of the unbiased gradient estimators[4]. We start with an illustrative problem and then follow the experimental setup established in (Maddison et al., 2016) to evaluate the methods on generative modeling and structured prediction tasks.

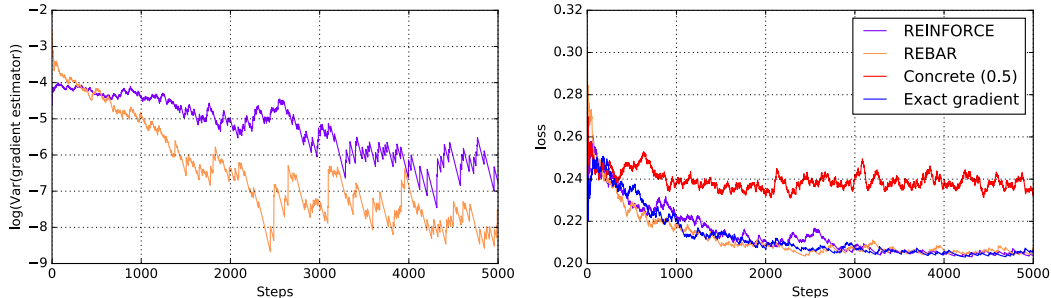

Figure 1: Log variance of the gradient estimator (left) and loss (right) for the toy problem with $t = 0.45$. Only the unbiased estimators converge to the correct answer. We indicate the temperature in parenthesis where relevant.

## 5.1 Toy problem

To illustrate the potential ill-effects of biased gradient estimators, we evaluated the methods on a simple toy problem. We wish to minimize $\mathbb{E}_{p(b)}[(b - t)^2]$, where $t \in (0, 1)$ is a continuous target value, and we have a single parameter controlling the Bernoulli distribution. Figure 1 shows the perils of biased gradient estimators. The optimal solution is deterministic (i.e., $p(b = 1) \in \{0, 1\}$), whereas the Concrete estimator converges to a stochastic one. All of the unbiased estimators correctly converge to the optimal loss, whereas the biased estimator fails to. For this simple problem, it is sufficient to reduce temperature of the relaxation to achieve an acceptable solution.

## 5.2 Learning sigmoid belief networks (SBNs)

Next, we trained SBNs on several standard benchmark tasks. We follow the setup established in (Maddison et al., 2016). We used the statically binarized MNIST digits from Salakhutdinov & Murray (2008) and a fixed binarization of the Omniglot character dataset. We used the standard splits into training, validation, and test sets. The network used several layers of 200 stochastic binary units interleaved with deterministic nonlinearities. In our experiments, we used either a linear deterministic layer (denoted linear) or 2 layers of 200 tanh units (denoted nonlinear).

### 5.2.1 Generative modeling on MNIST and Omniglot

For generative modeling, we maximized a single-sample variational lower bound on the log-likelihood. We performed amortized inference (Kingma & Welling, 2013; Rezende et al., 2014) with an inference network with similar architecture in the reverse direction. In particular, denoting the image by $x$ and the hidden layer stochastic activations by $b \sim q(b|x, \theta)$, we have

$$\log p(x|\theta) \geq \mathbb{E}_{q(b|x,\theta)} \left[ \log p(x, b|\theta) - \log q(b|x, \theta) \right],$$

which has the required form for REBAR.

To measure the variance of the gradient estimators, we follow a single optimization trajectory and use the same random numbers for all methods. This significantly reduces the variance in our measurements. We plot the log variance of the unbiased gradient estimators in Figure 2 for MNIST (Appendix Figure App.3 for Omniglot). REBAR produced the lowest variance across linear and nonlinear models for both tasks. The reduction in variance was especially large for the linear models. For the nonlinear model, REBAR (0.1) reduced variance at the beginning of training, but its performance degraded later in training. REBAR was able to adaptively change the temperature as optimization progressed and retained superior variance reduction. We also observed that SimpleMuProp was a surprisingly strong baseline that improved significantly over NVIL. It performed similarly to MuProp despite not explicitly using the gradient of $f$.

Generally, lower variance gradient estimates led to faster optimization of the objective and convergence to a better final value (Figure 3, Table 1, Appendix Figures App.2 and App.4). For the nonlinear model, the Concrete estimator underperformed optimizing the training objective in both tasks.

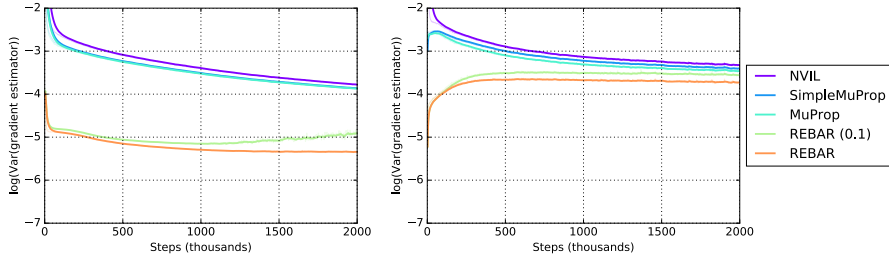

Figure 2: Log variance of the gradient estimator for the two layer linear model (left) and single layer nonlinear model (right) on the MNIST generative modeling task. All of the estimators are unbiased, so their variance is directly comparable. We estimated moments from exponential moving averages (with decay=0.999; we found that the results were robust to the exact value). The temperature is shown in parenthesis where relevant.

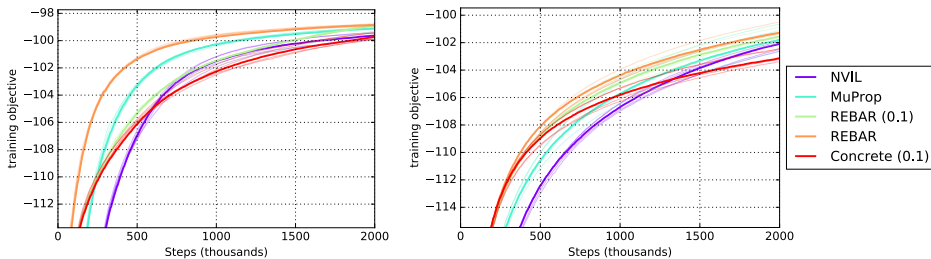

Figure 3: Training variational lower bound for the two layer linear model (left) and single layer nonlinear model (right) on the MNIST generative modeling task. We plot 5 trials over different random initializations for each method with the median trial highlighted. The temperature is shown in parenthesis where relevant.

Although our primary focus was optimization, for completeness, we include results on the test set in Appendix Table App.2 computed with a 100-sample lower bound Burda et al. (2015). Improvements on the training variational lower bound do not directly translate into improved test log-likelihood. Previous work (Maddison et al., 2016) showed that regularizing the inference network alone was sufficient to prevent overfitting. This led us to hypothesize that the overfitting results was primarily due to overfitting in the inference network ($q$). To test this, we trained a separate inference network on the validation and test sets, taking care not to affect the model parameters. This reduced overfitting (Appendix Figure App.5), but did not completely resolve the issue, suggesting that the generative and inference networks jointly overfit.

### 5.2.2 Structured prediction on MNIST

Structured prediction is a form of conditional density estimation that aims to model high dimensional observations given a context. We followed the structured prediction task described by Raiko et al. (2014), where we modeled the bottom half of an MNIST digit ($x$) conditional on the top half ($c$). The conditional generative network takes as input $c$ and passes it through an SBN. We optimized a single sample lower bound on the log-likelihood

$$\log p(x|c, \theta) \geq \mathop{\mathbb{E}}_{p(b|c,\theta)} \left[ \log p(x|b, \theta) \right].$$

We measured the log variance of the gradient estimator (Figure 4) and found that REBAR significantly reduced variance. In some configurations, MuProp excelled, especially with the single layer linear model where the first order expansion that MuProp uses is most accurate. Again, the training objective performance generally mirrored the reduction in variance of the gradient estimator (Figure 5, Table 1).

| MNIST gen. | NVIL | MuProp | REBAR (0.1) | REBAR | Concrete (0.1) |
|---|---|---|---|---|---|
| Linear 1 layer | −112.5 | −111.7 | −111.7 | −111.6 | **−111.3** |
| Linear 2 layer | −99.6 | −99.07 | −99 | **−98.8** | −99.62 |
| Nonlinear | −102.2 | −101.5 | −101.4 | **−101.1** | −102.8 |
| **Omniglot gen.** | | | | | |
| Linear 1 layer | −117.44 | −117.09 | −116.93 | **−116.83** | −117.23 |
| Linear 2 layer | −109.98 | −109.55 | **−109.12** | **−108.99** | −109.95 |
| Nonlinear | −110.4 | −109.58 | −109 | **−108.72** | −110.64 |
| **MNIST struct. pred.** | | | | | |
| Linear 1 layer | −69.17 | **−64.33** | −65.73 | −65.21 | −65.49 |
| Linear 2 layer | −68.87 | −63.69 | −65.5 | **−61.72** | −66.88 |
| Nonlinear | −54.08 | −47.6 | −47.302 | **−46.44** | −47.02 |

Table 1: Mean training variational lower bound over 5 trials with different random initializations. The standard error of the mean is given in the Appendix. We bolded the best performing method (up to standard error) for each task. We report trials using the best performing learning rate for each task.

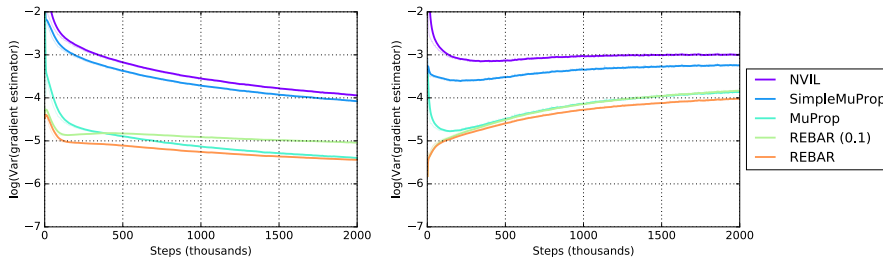

Figure 4: Log variance of the gradient estimator for the two layer linear model (left) and single layer nonlinear model (right) on the structured prediction task.

## 6 Discussion

Inspired by the Concrete relaxation, we introduced REBAR, a novel control variate for REINFORCE, and demonstrated that it greatly reduces the variance of the gradient estimator. We also showed that with a modification to the relaxation, REBAR and MuProp are closely related in the high temperature limit. Moreover, we showed that we can adapt the temperature online and that it further reduces variance.

Roeder et al. (2017) show that the reparameterization gradient includes a score function term which can adversely affect the gradient variance. Because the reparameterization gradient only enters the

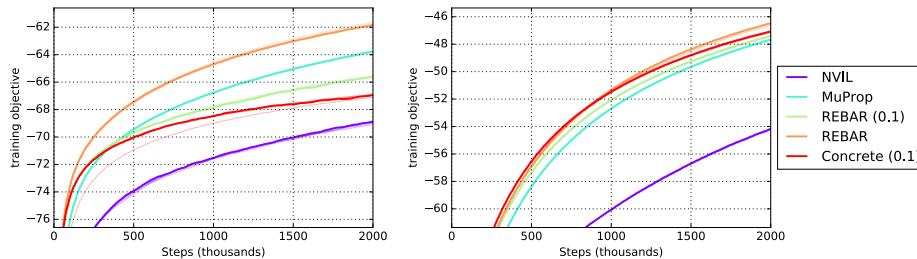

Figure 5: Training variational lower bound for the two layer linear model (left) and single layer nonlinear model (right) on the structured prediction task. We plot 5 trials over different random initializations for each method with the median trial highlighted.

REBAR estimator through differences of reparameterization gradients, we implicitly implement the recommendation from (Roeder et al., 2017).

When optimizing the relaxation temperature, we require the derivative with respect to $\lambda$ of the gradient of the parameters. Empirically, the temperature changes slowly relative to the parameters, so we might be able to amortize the cost of this operation over several parameter updates. We leave exploring these ideas to future work.

It would be natural to explore the extension to the multi-sample case (e.g., VIMCO (Mnih & Rezende, 2016)), to leverage the layered structure in our models using $Q$-functions, and to apply this approach to reinforcement learning.

### Acknowledgments

We thank Ben Poole and Eric Jang for helpful discussions and assistance replicating their results.

## Footnotes

[2]In this case, $c$ depends on the implicit observation in variational inference.

[3]$H(z) = 1$ if $z \geq 0$ and $H(z) = 0$ if $z < 0$.

[4]Both MuProp and REBAR require twice as much computation per step as NVIL and Concrete. To present comparable results with previous work, we plot our results in steps. However, to offer a fair comparison, NVIL should use two samples and thus reduce its variance by half (or $\log(2) \approx 0.69$ in our plots).

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
