[Supplementary Material · rebar_supp.pdf]

# Appendix

Figure App.1: Log variance of the gradient estimator for the single layer linear model on the MNIST generative modeling task (left) and on the structured prediction task (right).

Figure App.2: Training variational lower bound for the single layer linear model on the MNIST generative modeling task (left) and on the structured prediction task (right). We plot 5 trials over different random initializations for each method with the median trial highlighted.

Figure App.3: Log variance of the gradient estimator for the single layer linear model (left), the two layer linear model (middle), and the single layer nonlinear model (right) on the Omniglot generative modeling task.

## A   Control Variates

Suppose we want to estimate $\mathbb{E}_x[f(x)]$ for an arbitrary function $f$. The variance of the naive Monte Carlo estimator $\mathbb{E}_x[f(x)] \approx \frac{1}{k} \sum_i f(x^i)$, with $x^1, ..., x^n \sim p(x)$, can be reduced by introducing a control variate $g(x)$. In particular,

$$\mathbb{E}[f(x)] \approx \left( \frac{1}{k} \sum_i f(x^i) - \eta g(x^i) \right) + \eta \, \mathbb{E}[g(x)]]$$

is an unbiased estimator for any value of $\eta$. We can choose $\eta$ to minimize the variance of the estimator and it is straightforward to show that the optimal one is

$$\eta = \frac{\mathrm{Cov}(f, g)}{\mathrm{Var}(g)},$$

Figure App.4: Training variational lower bound for the single layer linear model (left), the two layer linear model (middle), and the single layer nonlinear model (right) on the Omniglot generative modeling task. We plot 5 trials over different random initializations for each method with the median trial highlighted.

| MNIST gen. | NVIL | MuProp | REBAR (0.1) | REBAR | Concrete (0.1) |
|---|---|---|---|---|---|
| Linear 1 layer | $-112.5 \pm 0.1$ | $-111.7 \pm 0.1$ | $-111.7 \pm 0.2$ | $-111.6 \pm 0.03$ | $\mathbf{-111.3 \pm 0.1}$ |
| Linear 2 layer | $-99.6 \pm 0.1$ | $-99.07 \pm 0.02$ | $-99 \pm 0.1$ | $\mathbf{-98.8 \pm 0.03}$ | $-99.62 \pm 0.09$ |
| Nonlinear | $-102.2 \pm 0.1$ | $-101.5 \pm 0.3$ | $-101.4 \pm 0.2$ | $\mathbf{-101.1 \pm 0.1}$ | $-102.8 \pm 0.2$ |
| **Omniglot gen.** | | | | | |
| Linear 1 layer | $-117.44 \pm 0.03$ | $-117.09 \pm 0.06$ | $-116.93 \pm 0.03$ | $\mathbf{-116.83 \pm 0.02}$ | $-117.23 \pm 0.04$ |
| Linear 2 layer | $-109.98 \pm 0.09$ | $-109.55 \pm 0.02$ | $\mathbf{-109.12 \pm 0.07}$ | $\mathbf{-108.99 \pm 0.06}$ | $-109.95 \pm 0.04$ |
| Nonlinear | $-110.4 \pm 0.2$ | $-109.58 \pm 0.09$ | $-109 \pm 0.1$ | $\mathbf{-108.72 \pm 0.06}$ | $-110.64 \pm 0.08$ |
| **MNIST struct. pred.** | | | | | |
| Linear 1 layer | $-69.15 \pm 0.02$ | $\mathbf{-64.31 \pm 0.01}$ | $-65.75 \pm 0.02$ | $-65.244 \pm 0.009$ | $-65.53 \pm 0.01$ |
| Linear 2 layer | $-68.88 \pm 0.04$ | $-63.68 \pm 0.02$ | $-65.525 \pm 0.004$ | $\mathbf{-61.74 \pm 0.02}$ | $-66.89 \pm 0.04$ |
| Nonlinear layer | $-54.01 \pm 0.03$ | $-47.58 \pm 0.04$ | $-47.34 \pm 0.02$ | $\mathbf{-46.44 \pm 0.03}$ | $-47.09 \pm 0.02$ |

Table App.1: Mean training variational lower bound over 5 trials with different random initializations and standard error of the mean. We report trials using the best performing learning rate for each task.

| MNIST gen. | NVIL | MuProp | REBAR (0.1) | REBAR | Concrete (0.1) |
|---|---|---|---|---|---|
| Linear 1 layer | $-108.35 \pm 0.06$ | $-108.03 \pm 0.07$ | $-107.74 \pm 0.09$ | $-107.65 \pm 0.08$ | $\mathbf{-107 \pm 0.1}$ |
| Linear 2 layer | $-96.54 \pm 0.04$ | $-96.07 \pm 0.05$ | $\mathbf{-95.47 \pm 0.08}$ | $-95.67 \pm 0.04$ | $-95.63 \pm 0.05$ |
| Nonlinear | $-100 \pm 0.1$ | $-100.66 \pm 0.08$ | $-100.48 \pm 0.09$ | $-100.69 \pm 0.08$ | $\mathbf{-99.54 \pm 0.06}$ |
| **Omniglot gen.** | | | | | |
| Linear 1 layer | $\mathbf{-117.59 \pm 0.04}$ | $-117.64 \pm 0.04$ | $-117.68 \pm 0.05$ | $\mathbf{-117.65 \pm 0.04}$ | $\mathbf{-117.65 \pm 0.05}$ |
| Linear 2 layer | $-111.43 \pm 0.04$ | $-111.22 \pm 0.04$ | $-110.97 \pm 0.07$ | $\mathbf{-110.83 \pm 0.06}$ | $-111.34 \pm 0.05$ |
| Nonlinear | $\mathbf{-116.57 \pm 0.08}$ | $-117.51 \pm 0.09$ | $-118.2 \pm 0.1$ | $-118.02 \pm 0.05$ | $\mathbf{-116.69 \pm 0.08}$ |
| **MNIST struct. pred.** | | | | | |
| Linear 1 layer | $-66.12 \pm 0.03$ | $-65.67 \pm 0.01$ | $-65.62 \pm 0.04$ | $-65.61 \pm 0.02$ | $\mathbf{-65.33 \pm 0.02}$ |
| Linear 2 layer | $-63.14 \pm 0.02$ | $-62.066 \pm 0.009$ | $-62.08 \pm 0.05$ | $\mathbf{-61.68 \pm 0.05}$ | $-61.91 \pm 0.03$ |
| Nonlinear | $-61.24 \pm 0.05$ | $-61.48 \pm 0.03$ | $-61.3 \pm 0.04$ | $-61.34 \pm 0.02$ | $\mathbf{-61.03 \pm 0.02}$ |

Table App.2: Mean test 100-sample variational lower bound on the log-likelihood Burda et al. (2015) over 5 random initializations with standard error of the mean. We report the best performing learning rate for each task based on the validation set.

and it reduces the variance of the estimator by $(1 - \rho(f, g)^2)$. So, if we can find a $g$ that is correlated with $f$, we can reduce the variance of the estimator. If we cannot compute $\mathbb{E}[g]$, we can use a low-variance estimator $\hat{g}$. This is the approach we take. Of course, we could define $\tilde{g} = g - \hat{g}$, which has zero mean, as the control variate, however, this obscures the interpretability of the control variate.

# B   Conditional marginalization for the control variate

In this section, we explain why the Monte Carlo gradient estimator derived from the left hand side of Eq. 2 has much lower variance than the Monte Carlo gradient estimator derived from the right hand side. This is because the left hand side can be seen as analytically performing a conditional

Figure App.5: Test 100-sample variational lower bound on the log-likelihood. We computed the bound using the inference network used during training (dash-dot line) and a separate inference network trained in parallel on the test set (solid line). The inference network trained on the test set gives a much tighter lower bound on the log-likelihood. However, we still observe overfitting.

marginalization

$$\mathop{\mathbb{E}}_{p(z)}\left[f(H(z))\frac{\partial}{\partial\theta}\log p(z)\right] = \mathop{\mathbb{E}}_{p(b)}\left[\mathop{\mathbb{E}}_{p(z|b)}\left[f(H(z))\frac{\partial}{\partial\theta}\log p(z)\right]\right] = \mathop{\mathbb{E}}_{p(b)}\left[f(b)\mathop{\mathbb{E}}_{p(z|b)}\left[\frac{\partial}{\partial\theta}\log p(z)\right]\right]$$

$$= \mathop{\mathbb{E}}_{p(b)}\left[f(b)\mathop{\mathbb{E}}_{p(z|b)}\left[\frac{\partial}{\partial\theta}(\log p(z|b) + \log p(b))\right]\right]$$

$$= \mathop{\mathbb{E}}_{p(b)}\left[f(b)\frac{\partial}{\partial\theta}\log p(b)\right],$$

where the equality in the second line follows from the fact that when $z \sim p(z|b)$, then $b = H(z)$, so $p(z) = p(z)\mathbf{1}(b = H(z)) = p(z)p(b|z) = p(z|b)p(b)$, where $\mathbf{1}(A)$ is the indicator function for event $A$.

A similar manipulation holds for the control variate

$$\mathop{\mathbb{E}}_{p(z)}\left[f(\sigma_\lambda(z))\frac{\partial}{\partial\theta}\log p(z)\right] = \mathop{\mathbb{E}}_{p(b)}\left[\mathop{\mathbb{E}}_{p(z|b)}\left[f(\sigma_\lambda(z))\frac{\partial}{\partial\theta}\log p(z)\right]\right]$$

$$= \mathop{\mathbb{E}}_{p(b)}\left[\mathop{\mathbb{E}}_{p(z|b)}\left[f(\sigma_\lambda(z))\frac{\partial}{\partial\theta}(\log p(z|b) + \log p(b))\right]\right]$$

$$= \mathop{\mathbb{E}}_{p(b)}\left[\frac{\partial}{\partial\theta}\mathop{\mathbb{E}}_{p(z|b)}[f(\sigma_\lambda(z))]\right] + \mathop{\mathbb{E}}_{p(b)}\left[\mathop{\mathbb{E}}_{p(z|b)}[f(\sigma_\lambda(z))]\frac{\partial}{\partial\theta}\log p(b)\right]$$

where again we used the fact that when $z \sim p(z|b)$, $p(z) = p(z|b)p(b)$.

## C   Reparameterizations for REBAR

In this section, we describe reparameterizations for $p(z)$ and $p(z|b)$ when $b$ is an categorical random variable, including the special case of $b$ binary. Let $b$ be a categorical random variable represented as a one-hot vector of 0s and 1s with probability $p_i > 0$ that $b_i = 1$. Note $p_i$ are normalized. Let $u_i \sim \mathrm{Uniform}(0, 1)$ be i.i.d. uniform random variables, and define the random vector $z$ by

$$z_i := g(u_i, p_i) := \log p_i - \log(-\log(u_i)) \tag{6}$$

We can parameterize $b$ as $H(z)$ where now $H : \mathbb{R}^n \to \{0, 1\}^n$ is the one-hot argmax function

$$H_i(z) = \begin{cases} 1 & \text{if } z_i \geq z_j \text{ for } j \neq i \\ 0 & \text{otherwise} \end{cases}$$

This is known as the Gumbel-Max trick (see Maddison et al. (2016); Jang et al. (2016) for a discussion of the literature), because the $z_i$ are Gumbel random variables with location parameters $\log p_i$. The Gumbel with location $\mu$ is a distribution with c.d.f. and density given respectively by

$$\Phi_\mu(z) = \exp(-\exp(-z + \mu))$$
$$\phi_\mu(z) = \exp(-z + \mu)\exp(-\exp(-z + \mu))$$

Thus, $p(z) = \prod_{i=1}^{n} \phi_{\log p_i}(z_i)$ and it has the reparameterization given by Eq. 6.

$\Phi$ has an important additive property in its location parameter that allows us to derive the distribution $p(z|b)$ and a reparameterization function $\tilde{g}(v, b, p)$. We have

$$\Phi_{\log(a+b)}(z) = \exp(-\exp(-z)(a+b)) = \Phi_{\log a}(z)\Phi_{\log b}(z)$$

Let $\mathbf{1}(A)$ be the indicator of event $A$, $b = H(z)$, and $k$ such that $b_k = 1$, then the joint is

$$
\begin{aligned}
p(b, z) &= \prod_{i=1}^{n} \phi_{\log p_i}(z_i)\mathbf{1}(z_k \geq z_i) \\
&= \frac{\Phi_{\log(1-p_k)}(z_k)}{\Phi_{\log(1-p_k)}(z_k)} \prod_{i=1}^{n} \phi_{\log p_i}(z_i)\mathbf{1}(z_k \geq z_i) \\
&= \phi_{\log p_k}(z_k)\Phi_{\log(1-p_k)}(z_k) \prod_{i \neq k} \frac{\phi_{\log p_i}(z_i)\mathbf{1}(z_k \geq z_i)}{\Phi_{\log p_i}(z_k)} \\
&= p_k \phi_0(z_k) \prod_{i \neq k} \frac{\phi_{\log p_i}(z_i)\mathbf{1}(z_k \geq z_i)}{\Phi_{\log p_i}(z_k)} \\
&= p(b)p(z|b)
\end{aligned}
$$

And so we see that $b$ has the correct categorical distribution, $z_k$ is a standard Gumbel random variable with location parameter 0, and $z_i$ for $i \neq k$ are Gumbel random variables with location parameter $\log p_i$ truncated at the value $z_k$. This derivation is a special case of the Top-Down construction of A* Sampling (Maddison et al., 2014). Maddison et al. (2014) derive a reparameterization for truncated Gumbel random variables, which we use to define the reparameterization function $\tilde{g}(v, b, p)$ of $p(z|b)$ via its $i$th component. Letting $v_i \sim \text{Uniform}(0, 1)$ and $k$ be such that $b_k = 1$:

$$
\tilde{g}_i(v, b, p) := \begin{cases} -\log(-\log v_k) & \text{if } i = k \\ -\log\left(-\frac{\log v_i}{p_i} - \log v_k\right) & \text{otherwise} \end{cases}
$$

Let $u \sim \text{Uniform}(0, 1)$, then the special case of binary $b \sim \text{Bernoulli}(\theta)$ reduces to

$$g(u, \theta) := \log \frac{\theta}{1-\theta} + \log \frac{u}{1-u}$$

and

$$
\tilde{g}(v, b, \theta) := \begin{cases} \log\left(\frac{v}{1-v}\frac{1}{1-\theta} + 1\right) & \text{if } b = 1 \\ -\log\left(\frac{v}{1-v}\frac{1}{\theta} + 1\right) & \text{if } b = 0 \end{cases}
$$

# D  An alternative view of REBAR

We can decompose the objective into a reparameterizable term and a residual

$$
\begin{aligned}
\frac{\partial}{\partial\theta} \mathop{\mathbb{E}}_{p(b)} [f(b)] &= \eta\frac{\partial}{\partial\theta} \mathop{\mathbb{E}}_{p(z)} [f(\sigma_\lambda(z)] + \frac{\partial}{\partial\theta} \mathop{\mathbb{E}}_{p(b)} \left[f(b) - \eta \mathop{\mathbb{E}}_{p(z|b)} [f(\sigma_\lambda(z))] \right] \\
&= \eta\frac{\partial}{\partial\theta} \mathop{\mathbb{E}}_{p(z)} [f(\sigma_\lambda(z)] \\
&\quad + \mathop{\mathbb{E}}_{p(b)} \left[ \left(f(b) - \eta \mathop{\mathbb{E}}_{p(z|b)} [f(\sigma_\lambda(z))]\right) \frac{\partial}{\partial\theta} \log p(b) - \eta\frac{\partial}{\partial\theta} \mathop{\mathbb{E}}_{p(z|b)} [f(\sigma_\lambda(z))] \right],
\end{aligned}
$$

where we expanded the second term similarly to REINFORCE. Importantly, the first and third terms are reparameterizable, so we can estimate them with low variance.

# E  Rethinking the relaxation and a connection to MuProp

Recall that the Concrete relaxation for binary variables is

$$h = H(z) \approx \sigma_\lambda(z) = \sigma\left(\frac{1}{\lambda}\log\frac{\theta}{1-\theta} + \frac{1}{\lambda}\log\frac{u}{1-u}\right).$$

Figure App.6: Graphical model depiction of $p(u_{1:2}|b_{1:2})$. Notably, $u_{1:2}$ are independent given $b_{1:2}$.

Since $\sigma_\lambda(z) \to \frac{1}{2}$ as $\lambda \to \infty$, this is clearly a poor approximation and will lead to an ineffective control variate. Alternatively, consider the relaxation

$$H(z) \approx \sigma\left(\frac{1}{\lambda}\frac{\lambda^2 + \lambda + 1}{\lambda + 1}\log\frac{\theta}{1-\theta} + \frac{1}{\lambda}\log\frac{u}{1-u}\right) = \sigma_\lambda(z_\lambda), \qquad (7)$$

where $z_\lambda = \frac{\lambda^2+\lambda+1}{\lambda+1}\log\frac{\theta}{1-\theta} + \log\frac{u}{1-u}$. As $\lambda \to \infty$, the relaxation converges to the mean, $\theta$, and still as $\lambda \to 0$, the relaxation becomes exact. Conveniently, we can think of this new relaxation as $\sigma_\lambda$ with a $\lambda$-dependent transformation of the parameters $\theta$.

Then, as $\lambda \to \infty$, the REBAR estimator converges to

$$\lim_{\lambda\to\infty}\left[[f(H(z_\lambda)) - \eta f(\sigma_\lambda(\tilde{z}_\lambda))]\frac{\partial}{\partial\theta}\log p(b)\bigg|_{b=H(z_\lambda)} + \eta\frac{\partial}{\partial\theta}f(\sigma_\lambda(z_\lambda)) - \eta\frac{\partial}{\partial\theta}f(\sigma_\lambda(\tilde{z}_\lambda))\right],$$

$$= [f(H(z)) - \eta f(\theta)]\frac{\partial}{\partial\theta}\log p(b)\bigg|_{b=H(z)},$$

which is MuProp without the linear term.

## F Multilayer stochastic network

The estimator for multilayer stochastic networks described in the main text requires $n$ passes through the network (where $n$ is the number of layers). We present two alternatives that do not require additional passes through the network.

Recall that we have multiple layers of stochastic units (i.e., $b = \{b_1, b_2, \ldots, b_n\}$) where $p(b)$ factorizes as

$$p(b_{1:n}) = p(b_1)p(b_2|b_1)\cdots p(b_n|b_{n-1}),$$

and similarly for the underlying Logistic random variables $p(z_{1:n})$ recalling that $b_i = H(z_i)$. We also defined a relaxed distribution $q(\tilde{z}_{1:n})$ where we replace the hard threshold function $H(z)$ with a continuous relaxation $\sigma_\lambda(z)$.

Now, we define a coupled distribution $p(z_{1:n}, \tilde{z}_{1:n})$ such that the marginals are $p(z_{1:n})$ and $q(\tilde{z}_{1:n})$ (see Appendix Figure App.6 for an example). Let $\theta_i$ by the parameters for the Bernoulli distribution defining $p(z_i|z_{1:i-1})$ and similarly for $\tilde{\theta}_i$. Then, $z_i = g(u_i, \theta_i) \sim p(z_i|z_{1:i-1})$ where $u_i$ is a vector of Uniform random variables. Similarly, let $\tilde{z}_i = g(u_i, \tilde{\theta}_i)$ where the randomness $u_i$ is shared. By construction, the marginals are $p(z_{1:n})$ and $q(\tilde{z}_{1:n})$. The key property of this coupled distribution is that both $q(\tilde{z}_{1:n})$ and $p(\tilde{z}_{1:n}|b_{1:n})$ are reparameterizable. By construction $\tilde{z}_{1:n}$ is a deterministic, differentiable function of $u_{1:n}$, so it suffices to understand $p(u_{1:n}|b_{1:n})$. By looking at the conditional dependence structure of this distribution (Appendix Figure App.6), it is clear that it factorizes (i.e., $p(u_{1:n}|b_{1:n}) = \prod_i p(u_i|b_{i-1:i})$) and as we showed in the single layer case, these individual factors are reparameterizable.

Similar to the single layer case (Appendix D), we can decompose the objective

$$\frac{\partial}{\partial \theta} \mathop{\mathbb{E}}_{p(b_{1:n})} [f(b_{1:n})] = \eta \frac{\partial}{\partial \theta} \mathop{\mathbb{E}}_{q(\tilde{z}_{1:n})} [f(\sigma_\lambda(\tilde{z}_{1:n}))]$$

$$+ \mathop{\mathbb{E}}_{p(b_{1:n})} \left[ \left( f(b_{1:n}) - \eta \mathop{\mathbb{E}}_{p(\tilde{z}_{1:n}|b_{1:n})} [f(\sigma_\lambda(\tilde{z}_{1:n}))] \right) \frac{\partial}{\partial \theta} \log p(b_{1:n}) - \eta \frac{\partial}{\partial \theta} \mathop{\mathbb{E}}_{p(\tilde{z}_{1:n}|b_{1:n})} [f(\sigma_\lambda(\tilde{z}_{1:n}))] \right].$$

where the first and third terms are reparameterizable. As $\lambda \to \infty$ this estimator converges to a multilayer version of SimpleMuProp. This is straightforward to implement and only requires two evaluations of $f$. Unfortunately, as the depth of the network increases, the continuous path will eventually diverge from the discrete path. We could compute additional passes through the network that transition every several layers. This would trade reduce variance at the cost of increased computation. We leave exploring these hybrid approaches to future work.

Alternatively, we can use the control variate

$$\mathop{\mathbb{E}}_{p(z_i|b_i, b_{i-1})} [Q(\sigma_\lambda(z_{i:n}), \theta, b_{1:i-1})] \frac{\partial}{\partial \theta} \log p(b_i|b_{i-1}),$$

where we train $Q$ to minimize[5]

$$\mathop{\mathbb{E}}_{p(b_{1:i})} \left[ \left( \mathop{\mathbb{E}}_{p(b_{i+1:n}|b_i)} [f(b)] - \mathop{\mathbb{E}}_{p(z_i|b_i)} [Q(\sigma_\lambda(z_i), \theta, b_{1:i-1})] \right)^2 \right].$$

This avoids the extra computation, and the $Q$-function could potentially learn to compensate for the continuous relaxation. We leave exploring this to future work.

## G  Implementation details

We used Adam (Kingma & Ba, 2014) with a constant learning rate from $\{3 \times 10^{-5}, 1 \times 10^{-4}, 3 \times 10^{-4}, 10^{-3}, 3 \times 10^{-3}\}$ for the linear models and from $\{3 \times 10^{-5}, 10^{-4}\}$ for the nonlinear models and decays $\beta_1 = 0.9, \beta_2 = 0.99999$[6]. Higher learning rates caused training to diverge. We used minibatches of 24 elements and optimized for 2 million steps. We centered the input to the inference network with the training data statistics. As in (Maddison et al., 2016), all binary variables took values in $\{-1, 1\}$. We initialized the bias of the output layer to the training data statistics as in (Burda et al., 2015). All of the unbiased estimators used input-dependent baselines as described in (Mnih & Gregor, 2014). We used a 10 times faster learning rate for the parameters of the baselines and control variate scalings.

Preliminary evaluations of the REBAR and Concrete estimators over a range of $\lambda$ found $\lambda = 0.1$ to perform well across tasks and configurations.

### G.1  MuProp

We found that the linear term in the MuProp baseline was detrimental for later layers, so we learned an additional scaling factor to modulate the linear terms. This reduced the variance of the MuProp learning signal beyond the algorithm described in (Gu et al., 2015).

### G.2  REBAR

We learned separate control variate scalings ($\eta$) for each parameter group (e.g., the weights in the first layer, the biases in first layer, etc.).

When computing the REBAR estimator, we leverage common random numbers to sample from $b, z$, and $z|b$. Recall that $z = g(u, \theta)$ where $u$ is a uniform random variable, $b = H(z)$, and

$z|b = \tilde{g}(v, b, \theta)$. The expectation in Eq. 4 is over $p(u, v)$ independently sampled from $\mathrm{Uniform}(0, 1)$, however, it is possible to draw $u$ and $v$ from a dependent joint distribution without changing the expected value. The key is that the first term of Eq. 4 depends only on $u$, the second on $v$ and $b$, the third on $u$, and the fourth on $v$ and $b$. So, if we sample $u$ uniformly and then generate $z$ and $b$, $z$ will be distributed according to $z|b$. We can also sample from $z|b$ by noting the point $u'$ where $g(u', \theta) = 0$ and sampling $v$ uniformly and then scaling it to the interval $[u', 1]$ if $b = 1$ or $[0, u']$ otherwise. So a choice of $u$ will determine a corresponding choice of $v$ which produces the same $z$. Importantly, $v$ and $b$ are independent after this generation procedure. We propose using this pair $(u, v)$ as the random numbers in the reparameterization trick.

### G.3 Concrete

A subtle point addressed in (Maddison et al., 2016) is that the objective optimized by the method we called Concrete and in Jang et al. (2016) is not a stochastic lower bound on the marginal log-likelihood. The results reported in (Maddison et al., 2016; Jang et al., 2016) were similar and REBAR is most similar to Jang et al. (2016), so we chose to compare against it. However, although the Concrete method does not optimize a lower bound, we emphasize that we evaluated a proper stochastic lower bound for all plots and numbers reported (including on the training set).

## Footnotes

[5] Note that this ignores the variance contribution from the reparameterizable terms. We leave evaluating this approach to future work.

[6] This large value of $\beta_2$ is crucial for online estimation of the temperature. Adam uses the same gradients to compute both the numerator and denominator for the update step. This can lead to biased updates, with greater bias for smaller $\beta_2$. This is especially severe when the distribution over gradient magnitudes is asymmetric and heavy tailed, as is the case here.