[Reviews · NeurIPS 2017]

Reviewer 1



Summary This paper proposes a control variate (CV) for the discrete distribution’s REINFORCE gradient estimator (RGE). The CV is based on the Concrete distribution (CD), a continuous relaxation of the discrete distribution that admits only biased Monte Carlo (MC) estimates of the discrete distribution’s gradient. Yet, using the CD as a CV results in an *unbiased* estimator for a discrete random variable’s (rv) path gradient as well as lower variance than the RGE (as expected). REBAR is derived by exploiting the REINFORCE estimator for the CD and by observing that given a discrete draw, the CD’s continuous parameter (z, here) can be marginalized out. REBAR has some nice connections to other estimators for discrete rv gradients, including MuProp. Moreover, the CD’s temperature parameter can be optimized on the fly since it is no longer crucial for trading off bias and variance (as it is when using the CD alone). Lastly, REBAR is applicable to both multi-level stochastic models as well as reinforcement learning. Experiments showing the improvements from de-biased estimators and lower estimator variance are included. Evaluation Method: I find REBAR to be a quite interesting and novel proposal. Using the CD as a CV is a nice idea and might have been worthy of publication on its own, but the additional idea of marginalizing out the CD’s continuous parameter (z) increases the worth of the contribution a step further. Moreover, REBAR is simple to implement so I can see it finding wide use everywhere REINFORCE is currently used. Discrete latent variables are not often used in stochastic NNs—-ostensibly b/c of the high variance that impedes training—-but REBAR may contribute here too, making discrete latent variables more competitive with continuous ones. I am hard pressed to find a deficiency with the proposed method. REBAR does not scale well to multiple stochastic layers, requiring a forward prop for each, but the authors propose workarounds for this in the appendix. Experiments: I find the experiments adequate to validate the method. The toy experiment examining bias is a little contrived, but it’s nice to see that the unbiased approach helps. A stochastic network experiment here would be much more impressive though. The following experiments show lower variance and higher ELBO when training. I appreciate the discussion of REBAR not directly resulting in improved held-out performance; this is something optimization-focused papers often leave out or fear to include. Presentation: I find the presentation of the method to be the paper’s most noteworthy deficiency. This is not all the authors’ fault for I recognize the page limit presents difficulties. However, I think the authors push too much information into the appendix, and this hurts the paper’s ability to stand alone. Without the appendix, REBAR’s derivation is quite hard to follow, I found. I would prefer the connection to MuProp be pushed into the Appendix and more explanation included in the REBAR derivation. Conclusions: I found this paper interesting and novel and expect it’ll have significant impact given the difficulties with REINFORCE and other MC gradient estimators for discrete rvs. Thus, I recommend its acceptance. My only complaint is that the method is hard to understand without the appendix, hurting the paper as a stand-alone read.

Reviewer 2



This paper introduces a control variate technique to reduce the variance of the REINFORCE gradient estimator for discrete latent variables. The method, called REBAR, is inspired by the Gumble-softmax/Concrete relaxations; however, in contrast to those, REBAR provides an unbiased gradient estimator. The paper shows a connection between REBAR and MuProp. The variance of the REBAR estimator is compared to state-of-the-art methods on sigmoid belief networks. The paper focuses on binary discrete latent variables. Overall, I found this is an interesting paper that addresses a relevant problem; namely, non-expensive low-variance gradient estimators for discrete latent variables. The writing quality is good, although there are some parts that weren't clear to me (see comments below). The connections to MuProp and Gumble-softmax/Concrete are clear. Please find below a list with detailed comments and concerns: - The paper states several times that p(z)=p(z|b)p(b). This is confusing, as p(z|b)p(b) should be the joint p(z,b). I think that the point is that the joint can be alternatively written as p(z,b)=p(b|z)p(z), where the first term is an indicator function, which takes value 1 if b=H(z) and zero otherwise, and that it motivates dropping this term. But being rigorous, the indicator function should be kept. So p(b|z)p(z)=p(z|b)p(b), and when b=H(z), then p(z)=p(z|b)p(b). I don't think the derivations in the paper are wrong, but this issue was confusing to me and should be clarified. - It is not clear to me how the key equation in the paper was obtained (the unnumbered equation between lines 98-99). The paper just reads "Putting the terms together, we arrive at", but this wasn't clear to me. The paper would definitely be improved by adding these details in the main text or in the Appendix. - In the same equation, the expectation w.r.t. p(u,v), where u, v ~ Uniform(0,1) is also misleading, because they're not independent. As Appendix 7.5 reads, "a choice of u will determine a corresponding choice of v which produces the same z", i.e., p(u,v)=p(u)p(v|u), with p(v|u) being deterministic. In other words, I don't see how "using this pair (u,v) as the random numbers" is a choice (as the paper suggests) rather than a mathematical consequence of the procedure. - The paper states that REBAR is equally applicable to the non-binary case (lines 44-45). I think the paper can be significantly improved by including the mathematical details in the Appendix to make this explicit. - In the experimental section, all figures show "steps" (of the variational procedure) in the x-axis. This should be replaced with wall-clock time to get a sense of the computational complexity of each approach. - In the experiments, it would be interesting to see a comparison with "Local expectation gradients" [Titsias & Lazaro-Gredilla]. - In the experimental section, do the authors have any intuition on why MuProp excels in the structured prediction task for Omniglot? (Figs. 6 and 7) - The procedure to optimize temperature (Sec 3.2) seems related to the procedure to optimize temperature in "Overdispersed black-box VI" [Ruiz et al.]; if that's the case, the connection can be pointed out. - The multilayer section (Sec 3.3) seems a Rao-Blackwellization procedure (see, e.g., [Ranganath et al.]). If that's the case, the authors should mention that. - In line 108, there seems to be a missing 1/lambda term. - In lines 143-144 and 193, there are missing parenthesis in the citations. - Consider renaming Appendix 7 as Appendix 1.